# Comparative Transcriptomics and Gene Knockout Reveal Virulence Factors of *Arthrinium phaeospermum* in *Bambusa pervariabilis × Dendrocalamopsis grandis*

**DOI:** 10.3390/jof7121001

**Published:** 2021-11-24

**Authors:** Xinmei Fang, Peng Yan, Mingmin Guan, Shan Han, Tianmin Qiao, Tiantian Lin, Tianhui Zhu, Shujiang Li

**Affiliations:** 1College of Forestry, Sichuan Agricultural University, Chengdu 611130, China; 2019104009@stu.sicau.edu.cn (X.F.); 201707355@stu.sicau.edu.cn (P.Y.); lilin3@stu.sicau.edu (M.G.); 13722@sicau.edu.cn (S.H.); tlin@sicau.edu.cn (T.L.); 10627@sicau.edu.cn (T.Z.); 2National Forestry and Grassland Administration Key Laboratory of Forest Resources Conservation and Ecological Safety on the Upper Reaches of the Yangtze River, Chengdu 611130, China; 11470@sicau.edu.cn

**Keywords:** *A. phaeospermum*, *B. pervariabilis × D. grandis*, differentially expressed genes, cutinase transcription factor 1 beta, gene function

## Abstract

*Arthrinium phaeospermum* can cause branch wilting of *Bambusa pervariabilis × Dendrocalamopsis grandis*, causing great economic losses and ecological damage. *A. phaeospermum* was sequenced in sterile deionized water (CK), rice tissue (T1) and *B. pervariabilis × D. grandis* (T2) fluid by RNA-Seq, and the function of *Ctf1β* 1 and *Ctf1β 2* was verified by gene knockout. There were 424, 471 and 396 differentially expressed genes between the T2 and CK, T2 and T1, and CK and T1 groups, respectively. Thirty DEGs had verified the change in expression by fluorescent quantitative PCR. Twenty-nine DEGs were the same as the expression level in RNA-Seq. In addition, ΔApCtf1β 1 and ΔApCtf1β 2 showed weaker virulence by gene knockout, and the complementary strains Ctf1β 1 and Ctf1β 2 showed the same virulence as the wild-type strains. Relative growth inhibition of ΔApCtf1β 1 and ΔApCtf1β was significantly decreased by 21.4% and 19.2%, respectively, by adding H_2_O_2_ compared to the estimates from the wild-type strain and decreased by 25% and 19.4%, respectively, by adding Congo red. The disease index of *B. pervariabilis × D. grandis* infected by two mutants was significantly lower than that of wild type. This suggested that *Ctf1β* genes are required for the stress response and virulence of *A. phaeospermum*.

## 1. Introduction

*Bambusa pervariabilis × Dendrocalamopsis**grandis* (hybrid bamboo), a kind of hybrid bamboo, is a vital renewable resource and is known as the “second forest” [1]. This hybrid bamboo has strong adaptability, cold resistance and drought resistance, a thick culm arm and uniform fiber length and is a good raw material for papermaking. Hybrid bamboo is dual-purpose and can be used for bamboo shoots and materials [2]. However, in recent years, hybrid bamboo shoot blight caused by *Arthrinium phaeospermum* has occurred in a large area of the Yangtze River basin, resulting in plant death and huge economic losses, threatening the construction of ecological barriers. *A. phaeospermum* is a plant pathogenic fungus distributed all over the world and a generalist with a broad range of host species [3]. In addition to *B. pervariabilis × D.*
*grandis* [4], its host plants also include cowpea, garden pea [5], sugarcane [6], *Phyllostachys prominens* [7,8], new olive [9], *Phyllostachys viridis* [10], etc.

Previous research on the shoot blight of *B. pervariabilis × D. grandis* caused by *A. phaeospermum* indicated that the AP protein toxin can damage the mitochondrial membrane of host *B. pervariabilis × D. grandis* shoots and inhibit respiration [11]. At the same time, the AP protein toxin affected the activities of defense enzymes such as superoxide dismutase, peroxidase, phenylalanine aminonyase, polyphenol oxidase, chitinase and β-1, 3 glucanase of *B. pervariabilis × D. grandis* [12], and its binding sites are proteins located on the cytoplasmic membrane of host plants. The toxin protein purified from the fermentation broth of *A. phaeospermum* by column chromatography and anion exchange chromatography showed strong virulence, different from the previously known pathogenic factors, such as arthrichitin and arthrinic acid, isolated from human skin diseases by Vijayakumar et al. [13] and Bloor [14]. Previous studies have found that the pathogenic active substances produced by the same fungus in different hosts and under different conditions are different [15,16]. In the biological control of this disease, we found that the chloramphenicol-resistant mutant of *Pseudomonas aeruginosa* ZB27 had a favorable effect on *B. pervariabilis × D. grandis* blight [17]. In addition to research on the occurrence and damage of the disease, the isolation mechanism of pathogenic toxins and the biocontrol mechanism of antagonistic microorganisms, we also tried to explore the infection mechanism of *A. phaeospermum* and the defense pathway of *B. pervariabilis × D. grandis* at the molecular level. Based on the whole genome information of *A. phaeospermum* [3], significant differences were found in the proteome level of *B. pervariabilis × D. grandis* after pathogen infection, among which the differential proteins were enriched mainly in the biological process and cell components [18]. There were also significant differences in the proteome and transcriptome levels of *B. pervariabilis × D. grandis* induced by inactivated protein toxin, among which the differentially abundant proteins were mainly enriched in biological processes and cell components. At the same time, the differentially expressed genes were involved mainly in lignin and phytoprotein synthesis, tetrapyrrole synthesis, redox reactions, photosynthesis, and other processes [19,20].

However, the interaction between host plants and fungal pathogens is well known to be essentially a coevolutionary mechanism. Pathogens invade host plants and then activate plant pathogen-associated molecular patterns-triggered immunity (PTI) due to the recognition of PAMPs of pathogenic fungi, including local cell wall reinforcement, induction of defense genes, secretion of chitinase, production of reactive oxygen species and production and release of antimicrobial compounds. To inhibit PTI, pathogens produce effectors to escape the recognition of immune receptors and induce the production of resistant proteins in plants to trigger ETI. Therefore, when pathogens interact with host plants, in addition to the metabolic changes in the host plants, the gene expression in the pathogens will also be significantly different [21,22,23,24].

In recent years, with the in-depth study of the interaction between pathogens and host plants and the rapid development of sequencing technology, RNA-seq has been widely used to study the mechanism of plant pathogen interactions. At present, RNA-seq in the study of plant pathogen interactions focuses mainly on screening host plant disease resistance genes and disease resistance metabolic pathways, which provides a theoretical basis for plant disease resistance mechanisms. For example, RNA-seq can be used in rice [25], *Arabidopsis thaliana* [26], tomato [27] and other host plants. However, RNA-seq is rarely used to study pathogens in the interaction process; in particular, the interaction between *A. phaeospermum* and *B. pervariabilis × D. grandis* has not been reported during host infection. It is necessary to screen some genes related to *A. phaeospermum* infection and pathogenicity by transcriptome analysis, but the specific functions of these candidate genes during infection are not clear. Gene knockout is an important means to study gene function. At present, the function of key genes in many fungi has been confirmed, including *Beauveria bassiana* [28], *Aspergillus niger* [29], and *Trichoderma harzianum* [30]. This paper sequenced and compared the transcriptome of fungi cultured on PDA medium, PDA medium containing 10% rice tissue fluid and PDA medium containing 10% *B. pervariabilis × D. grandis* tissue and verified the genes with significant differences by fluorescent quantitative PCR. Finally, we explored gene function by gene knockout. The purpose of this study was to provide reliable reference data for further study on the pathogenic mechanism of *A*. *phaeospermum* and the disease resistance mechanism of *B. pervariabilis × D. grandis*.

## 2. Materials and Methods

### 2.1. Meterials

Microorganism: *A. phaeospermum* was isolated by tissue isolation method [31] from diseased *B. pervariabilis × D. grandis*, the accession number of *A. phaeospermum* ITS in NCBI database is OK626768. The isolate was stored in China Forestry Culture Collection Center, numbered cfcc 86860 (http://www.cfcc-caf.org.cn/, accessed on 6 April 2007).

Plant tissue sample: One-year-old *B. pervariabilis × D. grandis* (bamboo seedlings were purchased from Shuyang Qichen Bamboo Seedling Co., Ltd., Suqian, China) were planted in the bamboo-growing areas of reclaimed farmland (103°01′ N, 29°54′ E) in Sichuan, China. The study area was at an altitude of 515.98 m with annual temperatures of 6.8 °C to 26.1 °C and annual precipitation of 1300–1700 mm. Pathogenicity test sample: One-year-old *B. pervariabilis × D. grandis* were planted in greenhouse of Chengdu, Sichuan, China at a temperature of 25 °C.

### 2.2. Methods

#### 2.2.1. Preparation of Transcriptome Sequencing Mycelia

The method of Andrade et al. [32] and Jiang et al. [33] was used to obtain sterile *B. pervariabilis × D. grandis* plant tissue fluid. Young shoots of *B. pervariabilis × D. grandis* were cut to 20 g, disinfected with 1.5% sodium hypochlorite for 1 min, disinfected with 75% alcohol for 3 min, washed with sterile deionized water three times, ground with 20 mL sterile deionized water and filtered with sterile gauze. The filtrate was centrifuged in a 20 mL centrifugal tube at 4 °C and 7000 rpm for 10 min. The supernatant was transferred to a new 20 mL centrifugal tube and centrifuged four times. Finally, the supernatant was removed by using a sterile 0.22-μm bacterial filter. The tissue liquid of *B. pervariabilis × D. grandis* was obtained and stored at −20 °C for subsequent experiments. The method of Andrade et al. [31] was also used to obtain sterile rice plant tissue fluid.

Mycelium treatment. Two milliliters of sterile deionized water, sterile rice tissue fluid and sterile *B. pervariabilis × D. grandis* were placed on PDA medium (Haibo Biotechnology Co., Ltd., Qingdao, China), spread evenly with a coater and placed on a clean worktable for future use. *A. phaeospermum* was inoculated into PDA medium coated with 2 mL sterile deionized water (CK), PDRA medium coated with 2 mL sterile rice tissue fluid (T1) and PDHA medium coated with 2 mL sterile *B. pervariabilis × D. grandis* tissue fluid (T2). Each treatment was repeated 3 times, cultured in a 25 °C incubator for 7 days and preserved.

#### 2.2.2. Extraction and Quality Detection of *A. phaeospermum* RNA

The phenol chloroform method [34] was used to extract total RNA from *A. phaeospermum* grown in sterile deionized water, sterile rice tissue fluid and sterile *B. pervariabilis × D. grandis* tissue fluid. Then, the genomic DNA was removed by DNase, RNase inhibitor (Takara, Dalian, China) and other reagents. Agarose gel electrophoresis, a Nanodrop microspectrophotometer (NANODROP, ThermoFisher Scientific-CN, Shanghai, China) and an Agilent 2100 bioanalyzer were used to test the total RNA integrity, purity and quality [35], respectively. If the ratio of 28S: 18S rRNA was 2:1, the RNA integrity was good. Among the total RNA purity tests, the A260/A280 ratio was optimal between 1.8 and 2.0.

#### 2.2.3. cDNA Preparation and Illumina Sequencing

PrimeScriptTM Double Strand cDNA Synthesis Kit (TaKaRa) was used to construct the cDNA library of *A. phaeospermum*. Poly(A) + RNA was isolated from aggregated RNA samples by using oligonucleotide (dT) beads [36]. Fragmentation buffer was added to interrupt the expression of mRNA: the short fragments were used as templates, and random hexamer primers were used to form the first strand of the cDNA. Then, the second strand of cDNA was synthesized by buffer, dNTPs, RNase H and DNA polymerase. The double-stranded cDNA was purified, and at the end of the repair, poly(A) and joints were added to establish a cDNA library [37]. The cDNA library was sequenced on an Illumina HiSeq2000 platform by Guangzhou Gene Denovo Biotechnology Co., Ltd. (Guangzhou, China).

#### 2.2.4. Sequence Data Filtering, De Novo Assembly and Contig Annotation

Before assembly, the low-quality sequences from raw reads were removed, including sequences with ambiguous bases (denoted with >10% ‘N’ in the sequence trace), low-quality reads (the rate of reads with a quality value ≤5 was more than 50%) and reads with adaptors. Transcriptome de novo assembly was carried out with the short read assembly program Trinity (Version: 2.1.1. Parameter: Kmer size = 31, Kmer cov = 7) [38]. TGICL software was used to cluster and remove redundant representative sequences. TGICL version number: tgicl-2.1: All parameters were default parameters, –*p* = 94 (minimum percent identity for overlaps), –l = 40 (minimum overlap length), and –v = 30 (maximum length of unmatched overhangs). We assembled nine samples (CK-1, CK-2, CK-3, T1-1, T1-2, T1-3, T2-1, T2-2 and T2-3) to obtain contigs and then used TGICL to cluster the contigs to remove redundancy. All annotations were performed on the merged contig set. To obtain GO information for the contig, first, the contig sequence was aligned to the protein database Nr (evalue < 0.00001) by blastx, and the protein with the highest sequence similarity to a given contig was obtained; thus, the Nr annotation information of the contig was obtained. Subsequently, according to Nr annotation information, contig GO annotation information was obtained using Blast2GO software [39]. The merged contig set function was annotated to the following databases by using BLAST+ (Version: 21:2.6.0+/TH2: 2.5.0+. Parameter: -evalue 10^−5^. http://www.ncbi.nlm.nih.gov/BLAST/, accessed on 10 November 2018) [40]: the Gene Ontology database (GO) [41] (http://www.geneontology.org/, accessed on 10 November 2018); the Kyoto Encyclopedia of Genes and Genomes (KEGG) [42] (http://www.genome.jp/kegg/, accessed on 10 November 2018); the Clusters of Protein Homology (KOG) [43] (http://www.ncbi.nlm.nih.gov/KOG/, accessed on 10 November 2018); the NCBI Non-Redundant Protein Sequence database (Nr) [44] (ftp://ftp.ncbi.nih.gov/blast/db/, accessed on 10 November 2018); and a manually annotated, nonredundant protein sequence database (Swiss-Prot) [45] (http://www.uniprot.org/, accessed on 10 November 2018). After obtaining GO annotations for each contig, we used WEGO software [46] to classify and count all contigs and understand the merged contig set function distribution characteristics in a macroscopic view.

#### 2.2.5. Analysis of Differentially Expressed Genes under Different Treatment Condition

The differential expression of pathogenic fungi in sterile deionized water (CK), sterile rice tissue fluid (T1) and sterile *B. pervariabilis × D. grandis* tissue fluid (T2) was analyzed. First, RSEM software [47] (Version 1.2.19) was used to align by calling bowtie2 to obtain the read count quantitatively. Then, the software package edgeR (http://www.bioconductor.org/packages/release/bioc/html/edgeR.html, accessed on 10 November 2018) based on R language was used for different analyses. Genes with significant differences in expression were defined as having a false discovery rate (FDR) < 0.05 and 2 log FC > 2. GO (Ashburner et al., 2000) [41] functional analysis and KEGG [42] pathway analysis of differential gene DEGs were carried out to search for virulence genes.

#### 2.2.6. Real-Time Fluorescence Quantitative PCR Verification

Firstly, we selected the genes with high differential multiples in several treatment groups, including up-regulated differential genes and down-regulated differential genes. In addition, 30 candidate genes were screened by combining the gene function annotation information in the database. We selected GAPDH and Actin as internal reference genes in RT-PCR analysis to determine whether RT-PCR with GAPDH and Actin as reference genes is consistent with transcriptional sequencing results, so as to verify the reliability of transcriptome sequencing data. Primer Premier 5.0 software was used to design fluorescence quantitative PCR primers using CDS regions of the internal reference gene GAPDH and 30 candidate genes (Table 1. There were 10 candidate genes in the Ck-vs-T1 treatment group, 10 candidate genes in the Ck-vs-T2 treatment group and 10 candidate genes in the T1-vs-T2 treatment group) as templates. The primer sequences of 30 candidate DEGs and internal reference genes are shown in Table 2. The length of the primer was 20–25 bp, and the length of the target amplification band was 100–200 bp. In this experiment, fluorescence quantitative polymerase chain reaction (CFX96-Real-Time System, Bio-Rad, Hercules, CA, USA) was used to quantify the changes in the expression of thirty candidate DEGs in sterile deionized water, rice tissue fluid and sterile *B. pervariabilis × D. grandis* tissue fluid. The qPCR system consisted of the following: 10 μL Mix (TSINGKE, Beijing, China), 7.4 μL ddH_2_O, F/R 0.8 μL, and 1 μL cDNA (CK/T2). The qPCR was as follows: 95 °C for 1 min, 95 °C for 30 s and 60 °C for 15 s. Forty cycles were repeated from the second step to the third step, and the dissolution curve was added at the final step. Each group of qPCR was repeated three times, and the average value was calculated. GAPDH was used as an internal reference gene to detect the expression changes of 30 candidate pathogenic genes in 3 different treatment groups. Data were analyzed by the 2^−^^△△Ct^ method [48].

#### 2.2.7. Screening of Antibiotic Types and Concentrations

*A. phaeospermum* was inoculated on PDA medium containing 100 µg/mL ampicillin, kanamycin, hygromycin and geneticin and cultured in a 25 °C incubator for 5–7 days. The growth status of *A. phaeospermum* was observed, and the antibiotics with the worst growth status were selected as selection markers. Concentrations of 0, 50, 100, 150, 200, 250, 300 and 350 µg/mL were set to select the optimal concentration of selected antibiotics.

#### 2.2.8. Construction of Knockout Vector

We used an improved split-marker method to identify gene function [49]. Taking the CDS region of cutinase transcription factor 1 beta 1 (*Ctf1β* 1) and cutinase transcription Factor 1 beta 2 (*Ctf1β* 2) gene as the center, 1300 bp upstream and 1300 bp downstream were selected as homologous arms to design primers to amplify homologous arms of *Ctf1β* from *A. phaeospermum* DNA. Meanwhile, primers were designed to amplify the hygromycin phosphotransferase gene (hph) from the pSilent-1 vector. (The primer sequence is shown in Table 3.) Amplification system: DNA 1 µL, *Ctf1β*-5-F/R (*Ctf1β*-3-F/R and hph-F/hph-R) 1/1 µL, 2 × TransTaq HiFi PCR SuperMix 25 µL, nuclease-free water 22 µL. Reaction procedure: 94 °C 5 min (94 °C 30 s, 55 °C 30 s, 72 °C 2 min) 34 cycles, 72 °C 10 min. *Ctf1β*-5, *Ctf1β*-3 and hph gene fragments were obtained. The fusion fragments were obtained by fusion PCR using LA Taq (Takara). The first round PCR amplification system: *Ctf1β*-5 3 µL, *Ctf1β*-3 3 µL, hph 6 µL, 10 × LA buffer 3 µL, dNTP Mix 1 µL, LA Taq 0.3 µL, ddH_2_O 13.7 µL. Reaction procedure: 94 °C for 3 min, (94 °C for 30 s, 55 °C for 1 min, 72 °C for 2 min), 34 cycles, 72 °C for 7 min. Second round PCR amplification system: The first round PCR product 2 µL, 10 × LA buffer 5 µL, dNTP mix 2 µL, LA Taq 0.5 µL, ddH_2_O 37.5 µL, *Ctf1β*-5-F/*Ctf1β*-3-R 1/1 µL. Reaction procedure: 94 °C 3 min (94 °C 30 s, 60 °C 30 s, 72 °C 3 min) 34 cycles, 72 °C 10 min. The fusion fragment *Ctf1β*-5-hph-*Ctf1β*-3 was obtained. Enzyme digestion system: 10 × QuickCut Green Buffer 5 μL, pCAMBIA0380 DNA 7 μL, ApaI 1 μL, Hind III 1 μL, ddH_2_O 36 μL. Procedure reaction: 37 °C for 5 min. We obtained vector pCAMBIA0380 linearized by ApaI and HindIII the fusion fragment containing the complementary sequences at ApaIand HindIII of vector pCAMBIA0380. The vector pCAMBIA0380 and fusion fragment *Ctf1β*-5-hph-*Ctf1β*-3 were ligated by clonExpress II one step cloning kit (Vazyme Biotech Co., Ltd., Nanjing, China). Linked reaction system: linearized pCAMBIA0380 5 μL. Fusion fragment 3 μL, 5 × CEIIBuffer 4 μL, Exnase 2 μL and ddH_2_O 6 μL. Procedure reaction: 37 °C for 30 min, and cooling at 4 °C for 5 min. The recombinant plasmid was transformed with DH5α (TransGen Biotech, Beijing, China), the plasmid was added to 50 µL DH5α, then ice bathed 25 min, 42 °C 30 s, ice bathed 2 min, 500 uL SOC liquid medium was been added in it, and shaken 200 rmp at 37 °C for 1 h. and detected by colony PCR.

#### 2.2.9. PEG-Mediated Protoplast Transformation

Protoplast preparation: Preparation of enzymatic hydrolysate: 0.2 g lysine, 0.5 g driselease, 20 mL 1.2 M KCl, magnetic stirring beads, stirring for 15 min, then 4 °C, 3500 rmp, acceleration of both up and down, centrifugation for 10 min. The supernatant was filtered through a bacterial filter to obtain the enzymatic hydrolysate. The mycelia were cultured for 2 days at 28 °C, and 180 rpm were filtered with single layer microcloth, washed with 25 mL 1.2 M KCl, added to the enzymatic hydrolysate, and cultured at 70 rpm and 30 °C for 9 h. The protoplasts were examined by microscopy, filtered by double-layer microscopy, washed with 25 mL of 1.2 M KCl, and centrifuged, and the supernatant was discarded. Then, 1 mL STC was added to the suspension for standby.

Genetic transformation: Then, 30 μL of the fusion fragment *Ctf1β*-5-hph-*Ctf1β*-3 was added to the 5 × 10^6^ protoplast and incubated at room temperature for 20 min. PEG was added three times, with an interval of 10 min each time and 400 μL was added each time. After standing at room temperature for 20 min, 10 mL TB3 liquid medium was added. The components of TB3 medium included yeast extract 3 g, acid hydrolyzed casein 3 g, sucrose 200 g and distilled water 1 L. Then, after culturing at 90 RMP and 25 °C for 10 h, 45 mL TB3 medium containing 350 µg/mL hyg antibiotic was added to the protoplast. The plate was poured after mixing, the culture was inverted at 25 °C for 3 days.

Transformant detection: Lysis buffer for microorganization to direct PCR (TaKaRa) was used to lyse the mycelium, and Ex Taq was used for detection. Hph-F/hph-R, (*Ctf1β*-3-hph)-F/R, (*Ctf1β*-5-hph)-F/R and *Ctf1β*-F/*Ctf1β*-R were used for internal detection.

#### 2.2.10. Phenotype Analysis of Transformant

To analyze the differences in vegetative growth among the strains, the morphology was observed, and the diameter of colonies was measured at 5 days after inoculation on PDA plates. Stress sensitivity assays were conducted on PDA plates supplemented with different agents: 2 mg/mL Congo red (CR), 2 mol/L NaCl, and 40 mmol/L H_2_O_2_ at 25 °C for 5 days. All assays were repeated three times, and all data were analyzed by one-way ANOVA and Duncan’s range test in SPSS 16.0 to measure specific differences between pairs of means. A *p* value of <0.05 was considered statistically significant.

#### 2.2.11. Pathogenicity Test

Twenty plants of two-year-old *B. pervariabilis × D. grandis* with uniform growth were selected. Five plants were randomly selected and sprayed on the upper eight shoots of each plant with the mycelial suspension of the wild-type strain. The remaining 15 plants were treated with the mycelial suspension of mutant ΔAP*Ctf1β* 1, the mycelial suspension of mutant ΔAP*Ctf1β* 2 and sterile water, with five plants per treatment. The samples were subjected to bagging moisturizing and sprayed once every 12 h, 3 times in total, using 3 independent replicates. The incidence was investigated 20 days after inoculation. The disease index was calculated as follows [50]. Disease grading standard: Grade 0: no wilt; Grade 1: less than 25% branches withered; Grade 2: 25–50% (including 25% and 50%) branches withered; Grade 3: 50–75% of the branches are dead (including 75%); Grade 4: more than 75% branches withered.
Disease index = [Σ (Disease grade × Number of diseased       branches)/(total branches) × The most serious disease grade] × 100.

#### 2.2.12. Complementation Test of Ctf1β Knockout

Taking the CDS region of hph gene as the center, 1300 bp upstream and 1300 bp down-stream were selected as homologous arms to design primers (the complementary sequence of the upstream primer of KanMx gene was added at the 5 ‘end of the downstream primer of the upstream homologous arm, the complementary sequence of the downstream primer of KanMx gene was added at the 5 ‘end of the upstream primer of the downstream homologous arm as shown in Table 3) to amplify homologous arms. The KanMx gene was amplified from PUG6 plasmid DNA as a screening marker gene, and kanMx-Ctf1β 1 and kanMx-Ctf1β 2 recombinant fragments were obtained by two rounds of fusion PCR. The gene knockout complement vectors KanMx-Ctf1β 1-5/KanMx-Ctf1β 1-3 and KanMx-Ctf1β 2-5/KanMx-Ctf1β 2-3 were constructed by using the improved spike marker method with reference to the method (Appendix A). The amplification system and reaction procedures of the first and second rounds of PCR were consistent with those in Section 2.2.8. Then, 30 µL each for KanMx-Ctf1β 1-5/KanMx + Ctf1β 1-3 and KanMx-Ctf1β 2-55/KanMx-Ctf1β 2-3 were added to the protoplast, and the fragments were transferred into the 5 × 10^6^ protoplasts according to the method in Section 2.2.9. The primers KanMx-Ctf1β 1-F/KanMx-Ctf1β 1-R and KanMx-Ctf1β 2-F/KanMx + Ctf1β 2-R were used for internal inspection. KanMx-Ctf1β 1-5-F/R and KanMx-Ctf1β 2-5-F/R were used for external inspection. Stress sensitivity assays were conducted on PDA plates supplemented with different agents: 2 mg/mL Congo red (CR), 2 mol/L NaCl, and 40 mmol/L H_2_O_2_ at 25 °C for 5 days to analyze the differences in vegetative growth among the strains. For the pathogenicity determination, twenty plants of two-year-old *B. pervariabilis × D. grandis* with uniform growth were selected. Five plants were randomly selected and sprayed on the upper eight shoots of each plant with the mycelial suspension of the wild-type strain. The remaining 15 plants were treated with the mycelial suspension of *Ctf1β* 1 complemented strain and *Ctf1β* 2 complemented strain, and sterile water, with five plants per treatment. The samples were subjected to bagging moisturizing and sprayed once every 12 h, 3 times in total, using 3 independent replicates. The incidence was investigated 20 days after inoculation. Refer to Section 2.2.11 for the calculation of disease index.

## 3. Results

### 3.1. Transcriptome Sequencing Results

#### 3.1.1. Growth of Fungi in Different Plant Tissue Culture Condition

The results showed that *A. phaeospermum* grew faster and better in sterile *B. pervariabilis × D. grandis* tissue liquid culture than in sterile deionized water and sterile rice tissue liquid culture. The growth curve of *A. phaeospermum* under three different culture conditions is shown in Figure 1. The results showed that the colony diameter cultured in the PDA medium containing *B. pervariabilis × D. grandis* tissue was 82.6 mm, which is significantly bigger than those of cultured in the medium containing rice tissue and sterile deionized water PDA medium.

#### 3.1.2. RNA Quality Detection

The quality of RNA was tested by agarose gel electrophoresis, NanoDrop spectrophotometry and Agilent 2100. The agarose gel electrophoresis results showed that the 28S:18S rRNA ratio of all the samples was 2:1, and RNA integrity was sound. In addition, the concentration of RNA in 9 samples was more than 500 ng/µL. In the total RNA purity test, the A260/A280 ratio of all samples was between 1.9 and 2.0. That is, the 9 RNA samples extracted had no DNA, impurity contamination or degradation and could meet the requirement of constructing libraries.

#### 3.1.3. Sequencing Data and DEG Statistics

The base percentage of Q30 in each sample after filtering was not less than 94.76%. The percentage of GC content was more than 56.46%. The reads’ lengths of 9 samples are 150 bp. The transcriptome sequencing results met the quality requirements of subsequent assembly analysis. The sequencing information of samples CK-1, CK-2, CK-3, T1-1, T1-2, T1-3, T2-1, T2-2 and T2-3 is shown in Table 4. We used short read alignment software Bowtie2 to align high-quality clean reads with reference contig sequences, which was obtained by transcriptome de novo assembly. Statistical results of comparisons between samples and reference contigs are shown in Table 5. In addition, we aligned the assembled *A. phaeospermum* transcriptome sequences with the *A. phaeospermum* genome. The alignment rates of 9 samples (CK-1, CK-2, CK-3, T1-1, T1-2, T1-3, T2-1, T2-2 and T2-3) were 85.00%, 86.08%, 85.11%, 84.81%, 84.17%, 79.07%, 83.10%, 81.61% and 83.97%, respectively (Appendix A). By Illumina sequencing and Trinity reassembly, the average continuous length was 2302 bp, and the N50 was 3326 bp. We assembled 30,620 contigs through Trinity and finally obtained 13,077 merged contigs through TGICL clustering and removing redundancy. Therefore, a total of 13,077 merged contigs were obtained from 9 samples. There were 12,845, 12,697 and 13,055 contigs in sterile deionized water and sterile *B. pervariabilis × D. grandis* tissue fluid, which accounted for 98.23%, 97.09% and 99.83% of the total contigs, respectively. Busco (version: 3.0.3, lineage dataset: Ascomycota_*odb9*) analysis results of sequence assembly quality are shown in Appendix A. Complete busco (including complete and single-copy BUSCOs and complete and duplicated BUSCOs) accounts for 92% of the total. All transcriptomic data of 9 samples of *A. phaeospermum* were deposited in the NCBI Sequence Reads Archive (SRA) under the accession numbers SRR9278662, SRR9278661, SRR9278664, SRR9278663, SRR9278658, SRR9278657, SRR9278660, SRR9278659 and SRR9278665. The assembled contigs have been published in the NCBI Transcriptome Shotgun Assembly (TSA) under the accession number GHWG00000000 (https://www.ncbi.nlm.nih.gov/nuccore/GHWG00000000.1/, accessed on 1 October 2019), including 12,861 contigs that came from the filters applied by TSA.

The contig sequences of *A. phaeospermum* under the three different treatment conditions were annotated and compared with SwissProt databases, Kyoto Encyclopedia of Genes and Genomes (KEGG) the Clusters of Protein Homology (KOG) and NCBI nonredundant (Nr) protein databases. The additional remonal redundant sequences obtained a total of 13,077 contigs. There were 10,155 contigs with annotated information in four major databases (Figure 2). Among these contigs, 3247 contigs were annotated in four databases at the same time. A total of 7271, 3714, 5904 and 9991 genes were annotated in the SwissPort, KEGG, KOG and NR databases, respectively. Based on the GO annotations, 3394 contigs were annotated into biological processes, including metabolic processes, cellular processes, single-organism processes and biological regulation. A total of 3111 contigs were annotated into cell components, in which contigs act mainly on cell parts, organelles, macromolecule complexes, cell membranes, etc. There were 2239 contigs annotated in molecular function. Most of them have catalytic activity, binding, transporter activity, structural molecular activity, molecular function regulator activity and other functions. The ten top species based on the Nr annotations are shown in Figure 3.

When comparing the DEGs of three different culture conditions in pairs, 4 DEGs were found to be upregulated and 367 were downregulated in CK-vs-T1 (Appendix A), 263 genes were upregulated and 161 were downregulated in CK-vs-T2 (Appendix A), 323 genes were upregulated and 148 were downregulated in T1-vs-T2 (Appendix A). The numbers of DEGs were similar between the CK-vs-T2 and T1-vs.-T2 comparisons, the number of DEGs in both comparison groups was large, and both contained cutinase transcription Factor 1 beta. The results showed a significant difference between the culture of the pathogen in the host plant *B. pervariabilis × D. grandis*. tissue medium and the culture of the pathogen in the nonhost plant sterile rice tissue medium or sterile deionized water. In contrast, the number of DEGs was the smallest in the CK-vs-T1 comparison, indicating a small difference between these two conditions. The differentially expressed genes between different groups under the three conditions are shown in a volcano plot (Figure 4).

### 3.2. KEGG and GO Annotation Analysis of DEGs

GO functional enrichment analysis was carried out for the differentially expressed genes of *A. phaeospermum* under three different treatment conditions. In the GO enrichment analysis of the CK and T1 treatment groups, there were no upregulated differentially expressed genes, while 91, 99 and 63 downregulated genes were annotated by biological processes, cell components and molecular functions, respectively (Figure 5A). In the GO enrichment analysis of the CK and T2 groups in the control group, 22, 10 and 33 upregulated genes and 36, 10 and 35 downregulated genes related to biological processes, cell components and molecular functions were obtained. The GO classification of the differentially expressed genes of *A. phaeospermum* in sterile deionized water and sterile *B. pervariabilis × D. grandis*. tissue fluid culture is shown in Figure 5B. In the GO enrichment analysis of the T1 and T2 treatment groups, 36, 10 and 53 upregulated genes and 47, 21 and 50 downregulated genes were obtained, respectively. The distribution of up- and downregulated genes in the GO function was basically consistent with the distribution of up- and downregulated genes in the CK and T2 treatment groups. The GO classification of differentially expressed genes of *A. phaeospermum* in sterile rice tissue fluid and sterile *B. pervariabilis × D. grandis*. tissue fluid culture is shown in Figure 5C. KEGG functional enrichment analysis was performed on the differentially expressed genes of *A. phaeospermum* treated with sterile deionized water, sterile rice tissue fluid and sterile hybrid *B. pervariabilis × D. grandis*. fluid. In the KEGG enrichment analysis of the CK and T1 treatment groups in the control group (Figure 6A), there were 144 DEGs in the KEGG pathway. One of the upregulated DEGs controls the metabolic pathway of oxidative phosphorylation. There were 143 downregulated DEGs, which mainly control RNA degradation, species longevity regulation and RNA transport. In the KEGG enrichment analysis of the CK and T2 treatment groups, the KEGG enrichment maps of CK in the control group and T2 in the treatment group are shown in Figure 6B. There were 43 DEGs and 25 upregulated DEGs. In the KEGG enrichment analysis of the T1 and T2 treatment groups, the KEGG enrichment maps of treatments T1 and T2 are shown in Figure 6C. The main enrichment processes were upregulation and downregulation DEGs, including 51 DEGs and 26 upregulation DEGs.

### 3.3. Verification by qRT–PCR

To verify the reliability of the RNA sequencing data, qRT–PCR was used to evaluate the expression under different culture conditions. When housekeeping genes GAPDH and Actin were used as internal reference genes, the expression trend of differential genes was the same and consistent with the results of transcriptome sequencing. We selected 10 DEGs from three different comparison groups to verify the change in expression (Figure 7). Among the 10 differentially expressed genes selected from the T1 and T2 treatment groups, 9 genes showed the same change trend as the result of transcriptome sequencing; only the change trend of *hydp* (hydrolase, partial) was inconsistent with the result of transcriptome sequencing, and the consistency rate was 90%. Among the 10 differentially expressed genes selected from the CK and T2 treatment groups, 10 genes showed the same change trend as transcriptome sequencing, with a consistent rate of 100%. Among the 10 differentially expressed genes selected from the CK and T1 treatment groups, all genes showed the same change trend as the result of transcriptome sequencing, with a consistent rate of 100%. Among these genes, the expression of *Ctf1β* in the T2 group was significantly higher than the expression of *Ctf1β* in the CK group, which may be related to *A. phaeospermum* infection of *B. pervariabilis × D. grandis*. needing to produce a large amount of cutinase to degrade cutin. In this case, *Ctf1β* is considered a gene related to *A. phaeospermum* virulence, it has further functional research value.

### 3.4. Gene Function Verification Results

#### 3.4.1. Construction of Knockout Vector and Genetic Transformation

Through the screening of antibiotic types and concentrations, when the concentration of hygromycin was 300 µg/mL or the concentration of genomycin was 50 µg/mL, the mycelium was found to almost not grow, and when the concentration of hygromycin reached 350 µg/mL or the concentration of genomycin was 100 µg/mL, the mycelium did not grow at all. Therefore, hygromycin and genomycin were selected as screening markers at concentrations of 350 µg/mL and 100 µg/mL, respectively. The kill curve for hygromycin and genomycin were in the Appendix A. The results showed that the growth of *A. phaeospermum* was inhibited by hygromycin. The higher the concentration of hygromycin, the stronger the inhibition. When the concentration of hygromycin was 350 µg/mL, the *A. phaeospermum* did not grow at all. Similarly, the growth of *A. phaeospermum* was inhibited by genomycin. The higher the concentration of genomycin, the stronger the inhibition. When the concentration of genomycin was 100 µg/mL, the *A. phaeospermum* did not grow at all. The hph, *Ctf1β1-5′*, *Ctf1β1-3′*, *Ctf1β2-5′* and *Ctf1β2-3′* fragments were obtained by PCR amplification. The knockout vectors of the *Ctf1β*1 and *Ctf1β*2 genes were constructed successfully by the improved split-marker method. From left to right were 15 K DNA marker (TransGen Biotech), recombinant plasmid DNA, empty plasmid DNA, recombinant plasmid double digested product and *Ctf1β* 1 and *Ctf1β* 2 knockout vector fusion fragments (Appendix A). The mycelium was hydrolyzed with 0.2 g lysine and 0.5 g drisealsel, and a large number of protoplasts were obtained, which were swollen and transparent at a dilution concentration of 10^7^/mL (Figure 8). After the fusion fragment was transferred into the protoplast and cultured in TB3 medium for 3 days, the colony diagram is as follows (Figure 9). A single colony was transferred to a PDA plate with a hygromycin concentration of 350 µg/mL, and several transformants were obtained after 4 days of culture. Then, “*hph*”, “*Ctf1β* 1”, “*Ctf1β* 2”, “uphph” and “downhph” fragments of wild-type, *Ctf1β* 1 and *Ctf1β* 2 deletion mutant strains were detected by PCR, as shown in Appendix A. The sequencing results were consistent with the *hph* gene sequence.

#### 3.4.2. Phenotypic Analysis of Transformant

The phenotypes of the mutants were significantly different from the phenotypes of the wild type (Figure 10). Based on the percent growth inhibition of the strains relative to unstressed controls, both mutants were significantly more sensitive to the oxidative stress of H_2_O_2_ than the control strains. The relative growth inhibition of ΔAp*Ctf1β* 1 and ΔAp*Ctf1β* 2 colonies was significantly decreased by 21.4% and 19.2%, respectively, by adding 40 mmol/L H_2_O_2_ compared to the estimates from the wild-type strain. Moreover, two disruption mutants exhibited significantly decreased tolerance to Congo red. ΔAp *Ctf1β* 1 and ΔAp *Ctf1β* 2 drastically decreased the colony growth areas by 25% and 19.4% at a concentration of 2 mg/mL Congo red, respectively. However, the hyperosmotic stress of 2 mol/L NaCl caused no significant differences in colony growth in either of the disruption mutants (Figure 11 and Figure 12). These results indicate that *Ctf1β 1* and *Ctf1β 2* play an important role in oxidative stress response and cell wall inhibitor stress response of *A. phaeospermum*.

#### 3.4.3. Pathogenicity Test of Transformant

To determine whether ΔAp*Ctf1β* 1 and ΔAp*Ctf1β* 2 are involved in pathogenicity, we performed a pathogenicity test on twigs and leaves by inoculating mycelial plugs of the wild type, ΔAp*Ctf1β* 1 deletion mutant and ΔAp*Ctf1β* 2 deletion mutant. Mild symptoms were found on the ΔAp*Ctf1β* 1 and ΔAp*Ctf1β* 2 mutant-infected twigs after culture in a greenhouse (temperature: 20 °C, humidity: 65%), while obvious symptoms on twigs and leaves of *B. pervariabilis × D. grandis*. were found in the wild-type strains (Figure 13). The statistical results of the disease index are shown in Figure 14. The disease index of *B. pervariabilis × D. grandis*. inoculated with sterile water did not change with the increase of time, and they were all 0. The disease index of *B. pervariabilis × D. grandis*. inoculated with wild type, mutant ΔAp*Ctf1β 1* and mutant ΔAp*Ctf1β*
*2* were increased significantly with the increase of inoculation time. After 25 days of inoculation, the disease index reached 86.25%, 60% and 60% respectively. At the same time, we found that the disease index of *B. pervariabilis × D. grandis*. inoculated with wild-type strain was significantly higher than that inoculated with mutant ΔAp*Ctf1β 1* strain and mutant ΔAp*Ctf1β 2* strain at the same time point. However, there was no significant difference in the disease index between mutant ΔAp*Ctf1β* 1 and ΔAp*Ctf1β* 2. (Figure 14). This result suggests that *Ctf1β* 1 and *Ctf1β* 2 play a key role in *A. phaeospermum* virulence.

#### 3.4.4. The Results of Complementation Test of Ctf1β Knockout

The knockout complement fusion fragments KanMx-Ctf1β 1-up/KanMx + Ctf1β 1-dowm and KanMx-Ctf1β 2-up/KanMx-Ctf1β 2-down were successfully constructed by the improved split-marker method. After the fusion fragment was transferred into the protoplast, it was cultured in TB3 medium for 3 days. A single colony was transferred to a PDA plate with a geneticin concentration of 100 µg/mL, and several transformants were obtained after 4 days of culture. Then, the “Ctf1β 1”, “KanMx-Ctf1β 1” and “KanMx-Ctf1β 1-up” fragments of the mutant strain and Ctf1β 1 complement mutant strains were detected by PCR, as shown in Appendix A. The sequencing results were consistent with the KanMx gene sequence.

### 3.5. Phenotypic Analysis and Pathogenicity Detection of Complementary Strains

There was no significant difference in phenotype between the two knockout complementary strains and the wild-type strain, and their sensitivity to oxidative stress of H_2_O_2_, tolerance to Congo red and salt tolerance to NaCl were similar (Figure 15). At the same time, the virulences of the two knockout complementary strains were consistent with the virulence of the wild-type, which was significantly enhanced compared with the mutant strain (Figure 16).

## 4. Discussion

Transcriptome sequencing data have become an integral component of modern genetics, genomics and evolutionary biology [51]. In this study, *A. phaeospermum* cultured on PDA medium, PDA medium containing rice and PDA medium containing *B. pervariabilis × D. grandis*. were sequenced by RNA-Seq. The results showed that the gene expression of pathogenic fungi was different among the three culture conditions. Similarly, transcriptome sequencing has been used to analyze the differential gene of the deformed mycelia, sclerotia and fruiting body in *Ophiocordyceps sinensis* [52], the differential gene of *Colletotrichum gloeosporioides* during the appressoria, quiescent and necrotrophic stages after infecting tomatoes [53], and the differential gene of *Magnaporthe oryzae* during spore germination and appressorium formation [54]. The genes encoding protein degradation and amino acid metabolism were significantly upregulated during the formation of the appressorium, while the expression of genes involved in protein synthesis was significantly decreased. In addition to genetic genes, the function and development process of organisms, the external environment is also one of the important factors affecting the growth and development of organisms and physiological functions. In the dynamic analysis of the transcriptome of maize seedlings under drought stress, the potential components of the abscisic acid signaling pathway were found to be significantly different between the two lines under drought stress [55]. Assembly analysis of the root transcriptome of Taxodium yasuensis under salt stress showed that there were 7959 differentially expressed genes under salt stress and nonsalt stress [56]. Therefore, during the growth and development of *A. phaeospermum* in different environments, the expression of genes closely related to the environment must be different. Compared with the fungus cultured on PDA medium and PDA medium containing nonhost plant rice, the expression of pathogenicity-related genes in the fungus cultured on PDA medium containing host plant *B. pervariabilis × D. grandis*. may be increased.

GO annotation analyses suggest that various metabolic relationships played a role under different culture conditions. Compared with fungi cultured on PDA, the differentially upregulated genes of pathogenic fungi cultured on *B. pervariabilis × D. grandis*. were involved mainly in cell biosynthesis, mitosis, nucleotide metabolism, organelle, cytoplasm and membrane formation, as well as translation factor activity, transmembrane transporter activity and hydrolase activity. KEGG pathway analysis suggested that the differentially upregulated genes were involved mainly in ribosome, endoplasmic reticulum protein processing, signal transduction of plant–pathogenic microorganism interaction, phosphoinositol signaling system, nitrogen metabolism, amino acid substitution thyme and metabolism of unsaturated fatty acids. It is intriguing that signal transduction in plant–pathogen interactions is a complex and necessary process in pathogen infection. From the beginning of contact recognition to the end of the susceptibility reaction, there are various signal transduction processes [57,58,59]. The differentially expressed genes were significantly enriched in the signal transduction pathway of plant-pathogenic microorganism interactions and the phosphoinositol signaling pathway. We speculated that the enriched genes in these two signaling pathways played an important role in the pathogenesis of *B. pervariabilis × D. grandis*.

The functional annotation showed that these differentially expressed genes included mainly the glycoside hydrolase family, secreted hydrolase, cutinase transcription Factor 1 beta and secreted hydrolase. Previous studies have found that the attachment of pathogenic fungi on the surface of host plants, the formation of infection structures, the penetration of host plants and colonization in host plants are the four key steps to establish pathogenicity. Pathogenic fungi infect host plants mainly by attaching to their surfaces through spores and germ tubes as a prerequisite [60,61,62,63]. This process occurs mainly through fungal enzymes secreted by pathogenic fungi to change the degree of plant surface adhesion to make it adhere. Gramineae *B. pervariabilis × D. grandis*. branches are composed mainly of cellulose and covered by cuticles, so the degrading cuticle, cellulose and cell wall of pathogens are usually closely related to pathogenicity [64,65]. The glycoside hydrolase family of hydrolases contains a variety of enzymes that are critical for lignocellulose degradation, which can degrade cellulose and hemicellulose, destroy plant cell walls and prevent lignification [36,66,67,68]. In addition to differentially expressed genes related to pathology, differentially expressed genes related to secondary metabolite synthesis were also significantly enriched. Some secondary metabolites of pathogenic fungi are toxic and pathogenic, of which mycotoxins are a major pathogenic factor in plant diseases [69]. For example, the secondary metabolite toxin of *Fusarium oxysporum* could cause cucumber fusarium wilt [70]. *Penicillium rosenbergii* produces a highly toxic secondary metabolite, PR toxin, which is a well-known isoprene mycotoxin [71]. Aflatoxin, a secondary metabolite of *Aspergillus flavus*, could cause many plant diseases, such as those of peanut, soybean and maize [72]. Therefore, the genes encoding hydrolase, lipase, toxic secondary metabolites and cutinase transcription factors among these differential genes were considered related to the pathogenicity of pathogens.

In this study, we observed that the symptoms of plant disease were different between wild-type and mutant strains by knocking out the *Ctf1β* 1 and *Ctf1β* 2 genes. At the same time, a knockout complementation experiment ruled out epigenetic changes caused by the mutagenicity of protoplast transformation itself. This result indicates that the virulence of the mutant strains ΔAP*Ctf1β* 1 and ΔAP*Ctf1β* 2 was lower than the virulence of wild-type *A. phaeospermum* to a certain extent. We speculate that the *Ctf1β* gene may be related to virulence. This speculation is inconsistent with the conclusion that the *Ctf1β* gene in the plant pathogen *Fusarium oxysporum* is not necessary for virulence. We speculate that the pathogen *F. oxysporum* is a soil-borne pathogen that enters the host plant mainly through the root lacking cuticle without breaking through the cuticle barrier on the plant surface [73]. However, the host plant of *A. phaeospermum* is a kind of *B. pervariabilis × D. grandis* with a thick/well-developed cuticle, which is a lipid layer on the outer surface of the cell wall of the plant surface and is divided into cuticle and wax [74]. In addition to the barrier function of water conservation and cleaning, it also has the complex defense function of promoting the overall development of plants and regulating the interaction between plants and pathogens [75,76,77]. To invade *B. pervariabilis × D. grandis*, the plant pathogen *A. phaeospermum* must secrete cuticle-degrading enzymes, including esterase, cutinase and lipase, which can catalyze the hydrolysis of ester bonds of lipoprotein, fat and wax to penetrate the outermost cuticle barrier of the host [78,79]. Cutinase is considered an important enzyme for fungi to penetrate the cuticle and infect plants. Cutinase plays an important role in attaching to the plant epidermis, cuticle invasion and signal generation. In addition to cutinase, secretory lipase is also a virulence factor of pathogenic fungi [80,81]. *Ctf1* was found to be the transcriptional activator of cutinase and lipase genes. An *F. oxysporum* strain lacking the function of the *Ctf1* gene was impaired in the induction of cutinase activity and the prediction of gene expression of cutinase and lipase, but the virulence of *Ctf1* to root pathogens was not necessary. However, *Ctf1α* of *Fusarium solani* F sp. pisi is a functional homologous gene of *Ctf1* that can control the expression of the cutinase gene and its own virulence. Although the pathogenicity of *F. solani* F sp. pisi was eliminated by *Ctf1α*, cutinase supplementation did not restore pathogenicity, which indicated that *Ctf1α* was involved in the regulation of other genes necessary for pathogenicity [82]. From the amino acid sequence, *Ctf1α* and *Ctf1β* have high homology with the proteins Fara and Farb of *Aspergillus nidulans*, and they also regulate the expression of genes related to lipid metabolism [83]. The outbreak of reactive oxygen species (ROS) in host plants is an important strategy for plants to inhibit pathogen infection. ROS are innate immune signals that can synthesize lignin and other phenolic polymers to block pathogen infection or act as second messengers to induce the expression of various plant defense-related genes and PAMP-triggered immunity (PTI) [84,85,86,87,88]. In this study, the loss of *Ctf1β* 1 and *Ctf1β* 2 gene function significantly increased the sensitivity of pathogens to extracellular oxidative stress (exposure to hydrogen peroxide). *Ctf1β* 1 and *Ctf1β* 2 have been proven to play an important role in the detoxification of reactive oxygen species (ROS). This result is consistent with the conclusion that the Bb*Ctf1 α* and Bb*Ctf1 β* deletion mutants of entomopathogenic *Beauveria bassiana* and the ccsge1 deletion mutants of the plant pathogen *Cytospora chrysosperma* are more sensitive to hydrogen peroxide [83,89]. The results showed that *Ctf1β* 1 and *Ctf1β* 2 were essential for oxidative stress and could inhibit the plant immune response. In addition, the loss of *Ctf1β* 1 and *Ctf1β* 2 gene function leads to the increased sensitivity of pathogens to cell wall disturbance stress (exposure to Congo red), consistent with the change in stress response sensitivity caused by the deletion of the *BbCtf1* α and *BbCtf1* β genes in *B. bassiana* [83]. These results demonstrate that *Ctf1β* is crucial in maintaining cell wall integrity.

## Figures and Tables

**Figure 1 jof-07-01001-f001:**
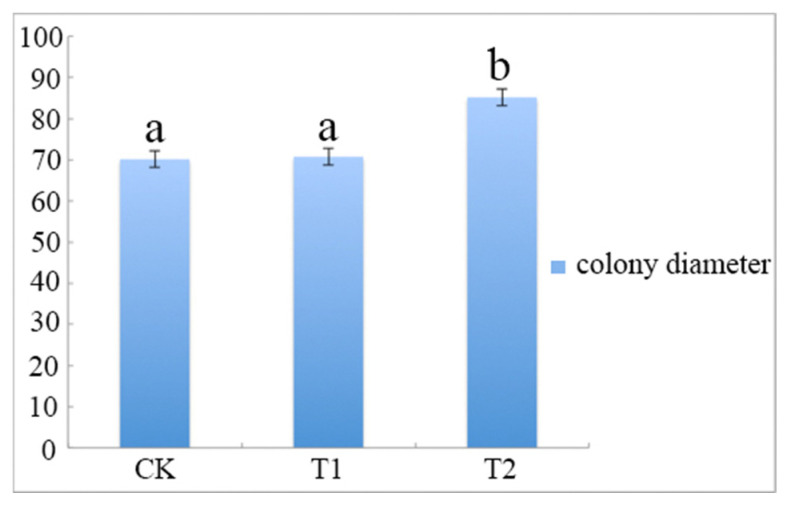
The colony diameter of *A. phaeospermum* after 7 days of culture under CK, T1 and T2. Note: All assays were repeated three times; the data were analyzed using one-way ANOVA and Duncan’s range test in SPSS 16.0. Different lowercase letters showed that there were significant differences in colony diameter under different treatment conditions (*p* ≤ 0.01).

**Figure 2 jof-07-01001-f002:**
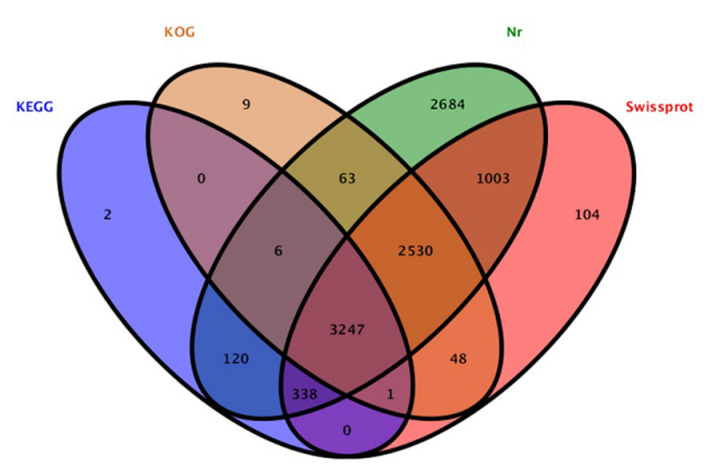
Annotation results of total genes in KEGG, KOG, Nr and Swissprot datasets.

**Figure 3 jof-07-01001-f003:**
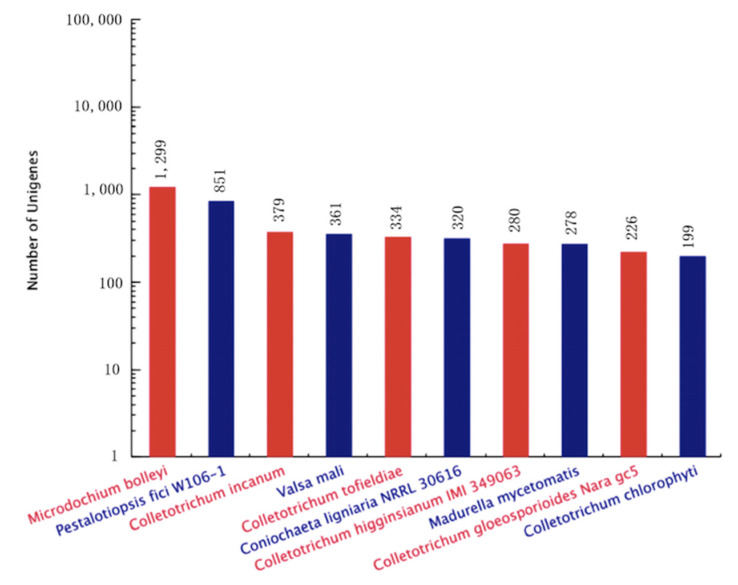
Statistical map of top ten species distribution.

**Figure 4 jof-07-01001-f004:**
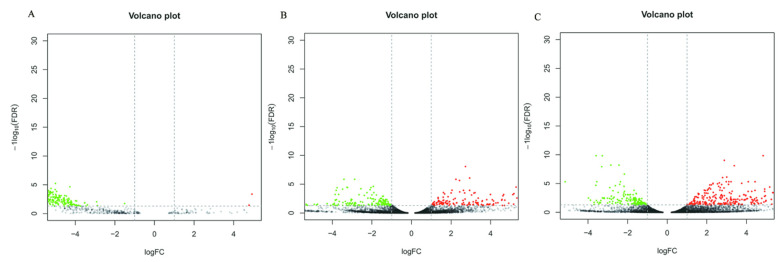
Differential Expressed Gene Volcano plot of CK vs. T1 (**A**), T1 vs. T2 (**B**) and CK vs. T2 (**C**) comparison groups. Note: Red represents genes that are upregulated, green represents genes that are downregulated, and black represents no difference. FDR < 0.05 and a difference multiple greater than 2 were used as the criteria for judging the difference in expression levels.

**Figure 5 jof-07-01001-f005:**
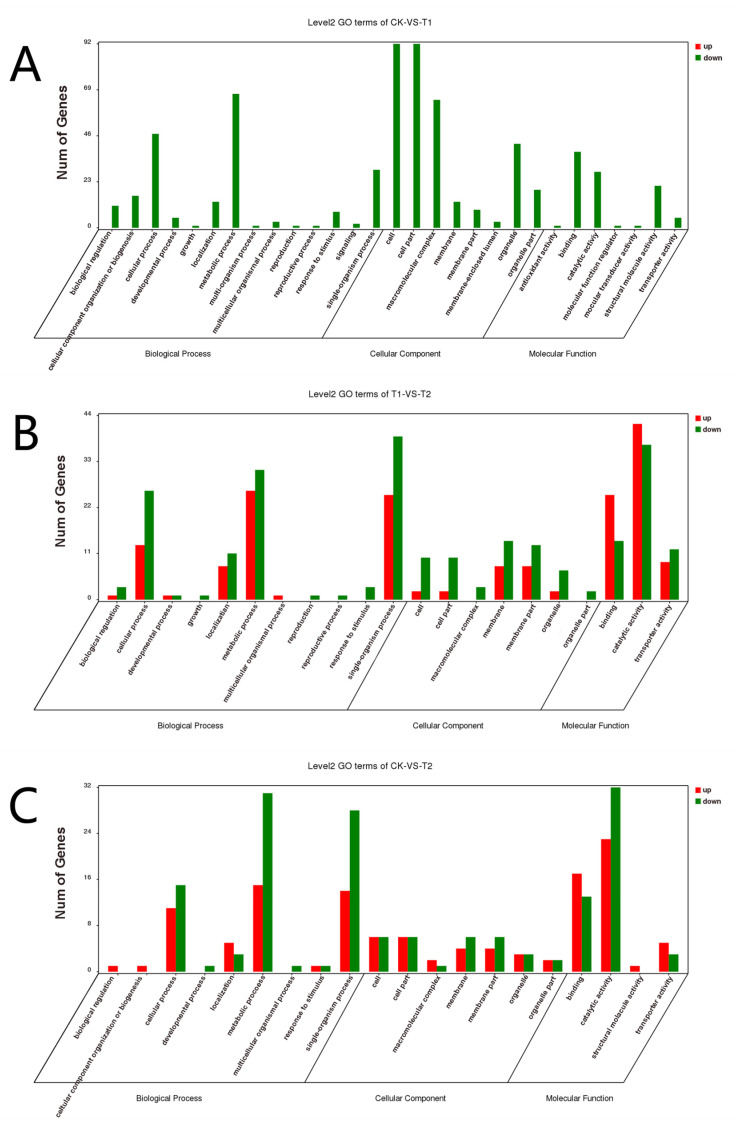
GO classification map of differentially expressed genes under different treatment conditions. (**A**) CK vs. T1; (**B**) CK vs. T2; (**C**) T1 vs. T2. Note: Red represents the up-regulated differentially expressed genes in the GO classification map, and green represents the down-regulated differentially expressed genes in the GO classification map.

**Figure 6 jof-07-01001-f006:**
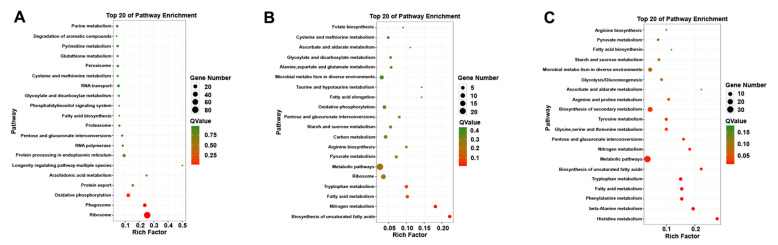
Summary of the KEGG enrichment map of different genes under different treatment conditions. (**A**) CK vs. T1; (**B**) CK vs. T2; (**C**) T1 vs. T2. Note: The size of the circle represents the number of differentially expressed genes KEGG enrichment map. The larger the circle, the more differentially expressed genes enriched in this pathway.

**Figure 7 jof-07-01001-f007:**
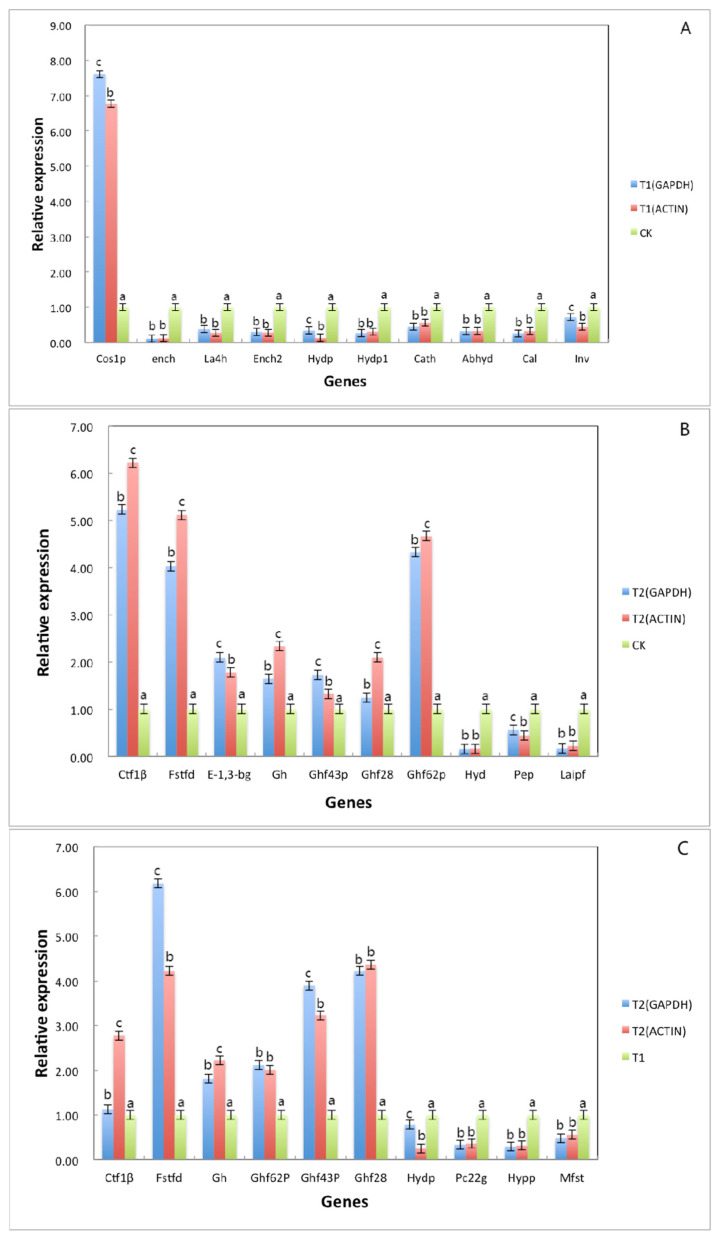
The differential gene expression under different treatment conditions was verified by fluorescence quantitative analysis. Note: (**A**) refers to the detection results of 10 differential gene expressions under two different culture conditions of sterile deionized water and sterile rice tissue fluid; (**B**) refers the detection results of 10 differential gene expressions under two different culture conditions of sterile deionized water and sterile tissue fluid of *B. pervariabilis × D. grandis*; (**C**) refers the detection results of 10 differential gene expressions under two different culture conditions of sterile rice tissue fluid and sterile *B. pervariabilis × D. grandis*. tissue fluid. All assays were repeated three times, the data were analyzed using one-way ANOVA and Duncan’s range test in SPSS 16.0. Different lowercase letters showed that there were significant differences in the expression level of the same gene under different treatment conditions. Blue represents the relative expression after normalization with GAPDH as the reference gene. Red represents the relative expression after normalization with actin as the reference gene (*p* ≤ 0.01).

**Figure 8 jof-07-01001-f008:**
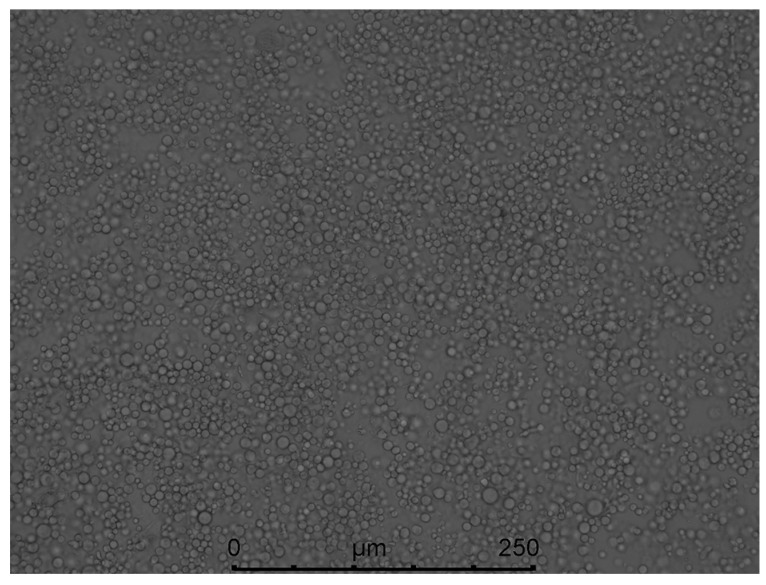
Morphology of *A. phaeospermum* protoplasts under 10 × 40 power optical microscope.

**Figure 9 jof-07-01001-f009:**
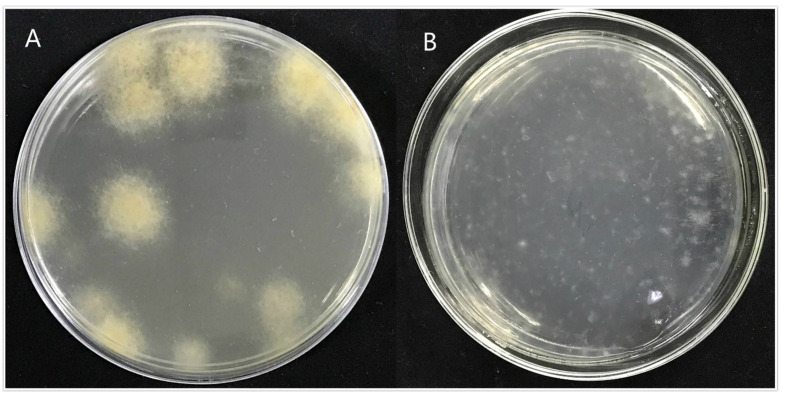
Colony diagram of transformant. Note: (**A**): Colony diagram of transformed strain ΔAp*Ctf1 β* after 4 days of incubation at 25. (**B**): Colony diagram of transformed strain ΔAp*Ctf1 β* after one day of incubation at 25.

**Figure 10 jof-07-01001-f010:**
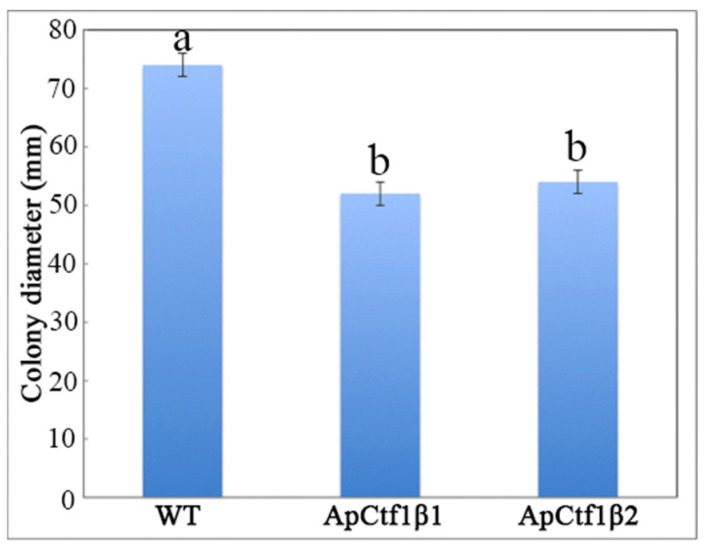
WT, ApCtf1 β 1 and ApCtf1 β 2 colony diameter after 7 days of culture at 25 °C. Note: WT, ApCtf1β1 and ApCtf1β2 show the colony diameters of the wild-type, Ctf1β1 deletion mutant and Ctf1β2 deletion mutant of *A. phaeospermum* after culturing for 7 days, respectively. All assays were repeated three times, the data were analyzed using one-way ANOVA and Duncan’s range test in SPSS 16.0. Different lowercase letters showed significant differences in colony diameter of strains (*p* ≤ 0.01).

**Figure 11 jof-07-01001-f011:**
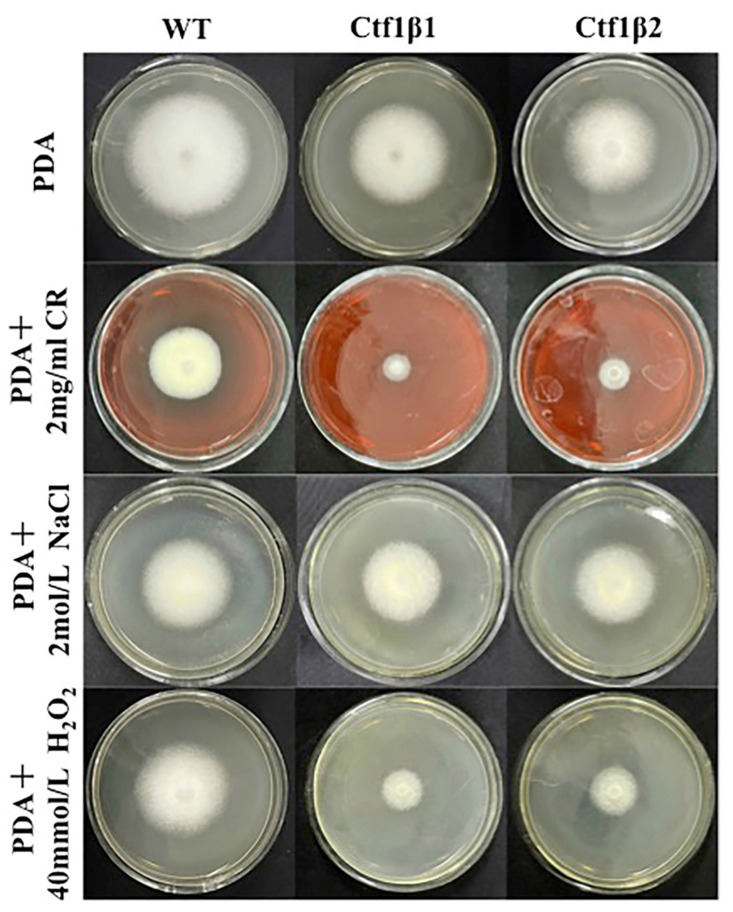
Comparison of the colony morphology and stress tolerance of wild-type, ΔAp*Ctf1β1* and ΔAp*Ctf1β2* strains. Note: The wild-type, *Ctf1β1* deletion and *Ctf1β2* deletion strains were inoculated on PDA media or PDA media appended with various stressors and cultured at 25 °C in darkness for 5 days.

**Figure 12 jof-07-01001-f012:**
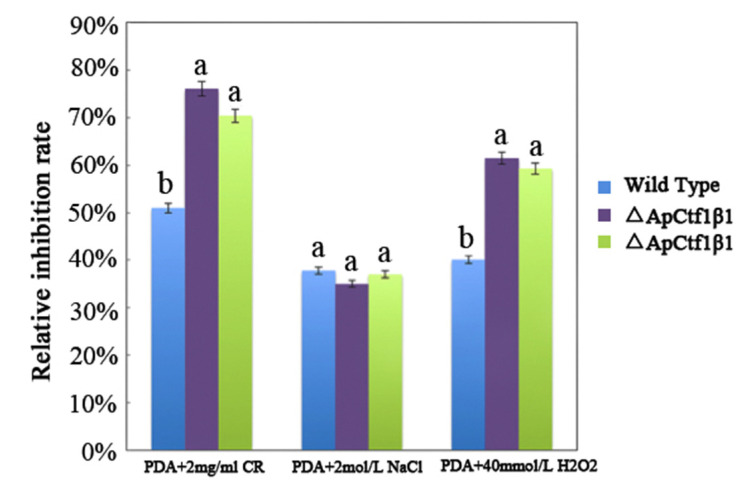
The bar chart showed the relative inhibition rate of wild type, ΔAp*Ctf1β 1* and ΔAp*Ctf1β 2* strains after CR, NaCl and H_2_O_2_ stress for 5 days, respectively. Note: The datasets were calculated from the image in Figure 11. The error bars represent the standard deviations based on three independent biological replicates with three technical replicates each. The relative inhibition rate of ΔAp*Ctf1β* 1 and ΔAp*Ctf1β 2* were significantly different from that of wild type after CR and H_2_O_2_ stress. The relative inhibition rate of ΔAp*Ctf1β 1* and ΔAp*Ctf1β 2* were consistent with that of wild type after NaCl stress. All assays were repeated three times, the data were analyzed using one-way ANOVA and Duncan’s range test in SPSS 16.0. Different lowercase letters showed that the relative inhibition rates of Congo red, NaCl and H_2_O_2_ on the mycelial growth of different strains were significantly different (*p* ≤ 0.01).

**Figure 13 jof-07-01001-f013:**
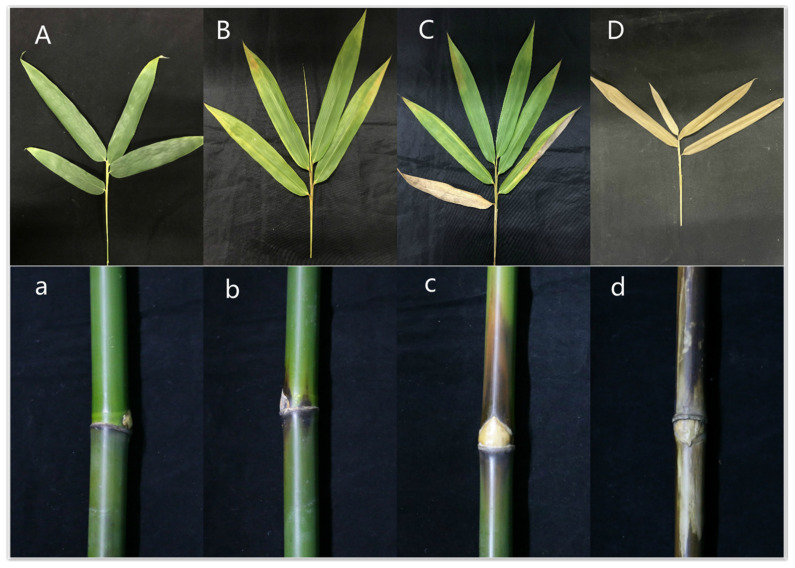
Symptoms of branches and leaves of plants infected with sterile water, ΔAp*Ctf1β 1* and ΔAp*Ctf1β 2* and wild-type strains after 25 days. Note: (**A**,**a**) are the symptoms of *B. pervariabilis × D. grandis* leaves and branches inoculated with sterile water for 25 days, respectively. (**B**,**b**) are the symptoms of *B. pervariabilis × D. grandis l*eaves and branches inoculated with ΔAp*Ctf1β 1* strain for 25 days, respectively. (**C**,**c**) are the symptoms of *B. pervariabilis × D. grandis l*eaves and branches inoculated with ΔAp*Ctf1β 2* strain for 25 days, respectively. (**D**,**d**) are the symptoms of *B. pervariabilis × D. grandis l*eaves and branches inoculated with wild type strain for 25 days, respectively.

**Figure 14 jof-07-01001-f014:**
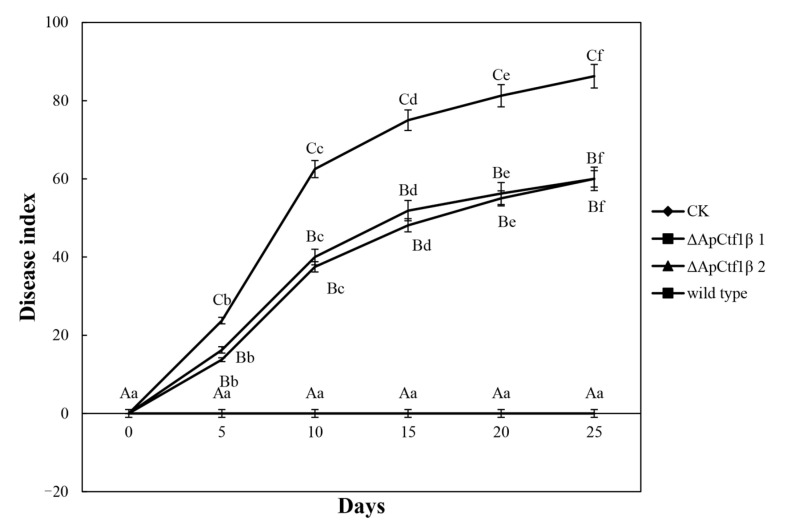
Dynamic changes in the disease index of *B. pervariabilis × D. grandis*. infected by different strains. Note: CK, ΔAp*Ctf1β1*, ΔAp*Ctf1β2* and wild type represent the changes in the disease index of *B. pervariabilis × D. grandis*. inoculated with sterile deionized water, *Ctf1β1* deletion mutant, *Ctf1β2* deletion mutant and wild-type *A. phaeospermum,* respectively. All assays were repeated three times, the data were analyzed using one-way ANOVA and Duncan’s range test in SPSS 16.0. Different capital letters indicate that the disease index of different strains in the same period is significantly different, and different lowercase letters indicate that the disease index of the same strain in different periods is significantly different (*p* ≤ 0.01).

**Figure 15 jof-07-01001-f015:**
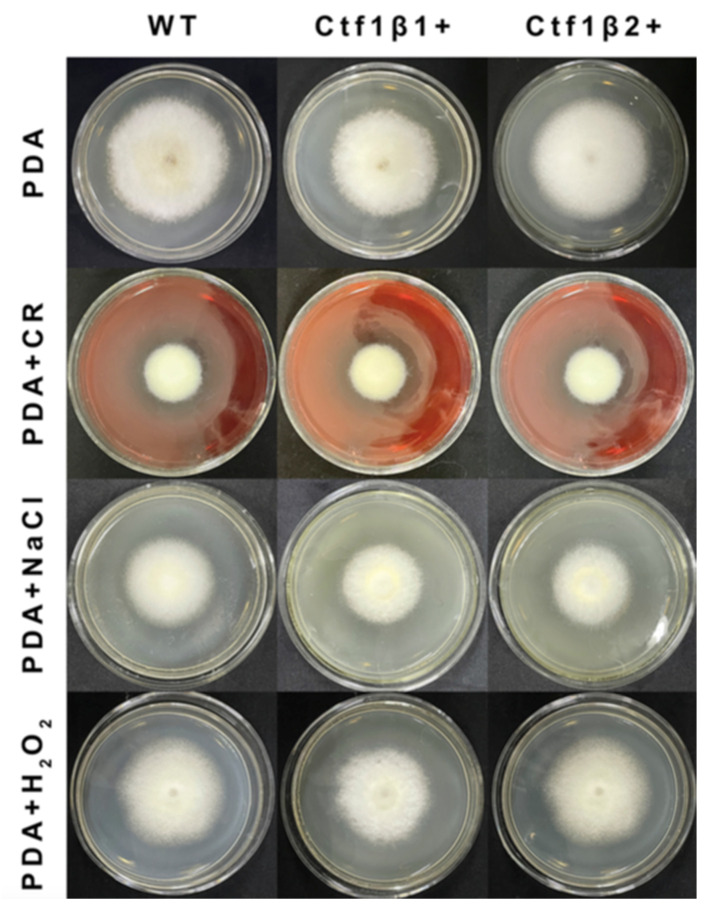
Comparison of phenotypic tolerance of the wild-type, Ctf1β 1 complemented strain and Ctf1β 2 complemented strain. Note: Wild-type, Ctf1β 1 complemented strain and Ctf1β 2 complemented strain were inoculated on PDA media or PDA media appended with various stressors and cultured at 25 °C in darkness for 5 days.

**Figure 16 jof-07-01001-f016:**
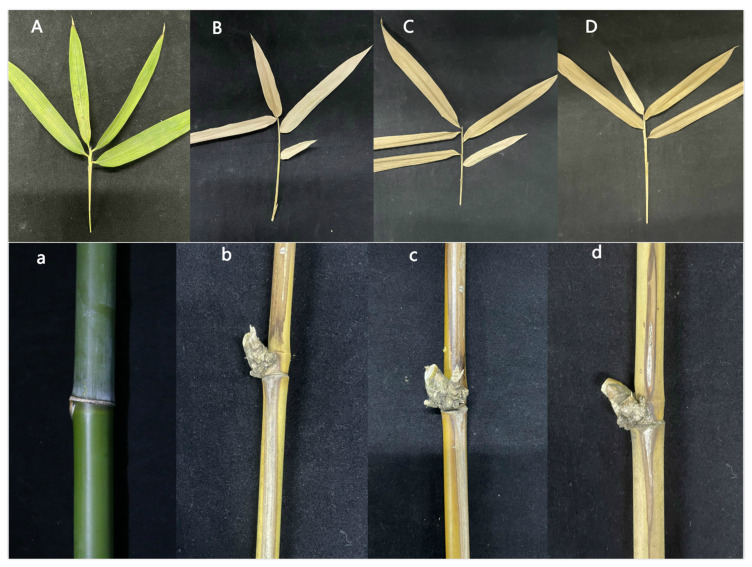
Symptoms of branches and leaves of plants infected with sterile water, Ctf1β 1 complemented strain, Ctf1β 2 complemented strain and wild-type strains after 25 days. Note: (**A**,**a**) are the symptoms of *B. pervariabilis × D. grandis* leaves and branches inoculated with sterile water for 25 days, respectively. (**B**,**b**) are the symptoms of *B. pervariabilis × D. grandis l*eaves and branches inoculated with Ctf1β 1 complemented strain for 25 days, respectively. (**C**,**c**) are the symptoms of *B. pervariabilis × D. grandis l*eaves and branches inoculated with Ctf1β 2 complemented strain for 25 days, respectively. (**D**,**d**) are the symptoms of *B. pervariabilis × D. grandis l*eaves and branches inoculated with wild type strain for 25 days, respectively.

**Table 1 jof-07-01001-t001:** Thirty Candidate validation genes.

Comparison Group	Gene Name	Log2 Ratio	Description
**CK-VS-T1**	*Cos1P*	9.956230937	cytochrome oxidase subunit 1, partial [*Brevilegnia unisperma*]
	*Ench*	−11.38168714	endochitinase [*Amphiamblys* sp. WSBS2006]
	*La4H*	−5.228792158	leukotriene A-4 hydrolase/aminopeptidase
	*Ench2*	−10.25223177	endochitinase [*Amphiamblys* sp. WSBS2006]
	*HydP*	−5.849553363	hydrolase, partial [*Mucor ambiguus*]
	*HydP1*	−5.423936695	hydrolase, partial [*Mucor ambiguus*]
	*CatH*	−5.21899521	cathepsin H [*Saprolegnia diclina* VS20]
	*abhyd*	−5.16654726	alpha/beta hydrolase [*Magnaporthe oryzae* 70–15]
	*Cal*	−5.003422577	Calmodulin [*Phytophthora nicotianae*]
	*Inv*	−4.361126768	Inversin [*Phytophthora nicotianae*]
**CK-VS-T2**	*Ctf1b*	1.16472	Cutinase transcription factor 1 beta
	*Fstfd*	4.66024	Fungal specific transcription factor domain
	*E-1,3-bg*	1.60198	Endo-1,3(4)-beta-glucanase
	*Gh*	1.926701059	glycoside hydrolase [*Stagonospora* sp. SRC1lsM3a]
	*Ghf43P*	2.046533818	glycoside hydrolase family 43 protein [*Trichoderma atroviride*]
	*Ghf28*	3.242824341	glycosyl hydrolase family 28 [*Colletotrichum incanum*]
	*Ghf62P*	3.263709993	glycoside hydrolase family 62 protein [*Schizophyllum commune*]
	*Hyd*	−9.113389998	hydrophobin [*Trichoderma atroviride* IMI 206040]
	*Pep*	−6.728250664	permease, partial [*Mucor ambiguus*]
	*LaipF*	−6.542070296	Low affinity iron permease, Fet4 [*Sporothrix insectorum*]
**T1-VS-T2**	*Ctf1b*	1.143966714	Cutinase transcription factor 1 beta
	*Fstfd*	5.178921385	Fungal specific transcription factor domain
	*Gh*	1.674955343	glycoside hydrolase [*Stagonospora* sp. SRC1lsM3a]
	*Ghf62P*	3.240379576	glycoside hydrolase family 62 protein [*Schizophyllum commune*]
	*Ghf43P*	2.798082794	glycoside hydrolase family 43 protein [*Trichoderma atroviride*]
	*Ghf28*	3.368988569	glycosyl hydrolase family 28 [*Colletotrichum incanum*]
	*HydP*	9.935115927	hydrolase, partial [*Mucor ambiguus*]
	*Pc22g*	−3.307249137	Pc22g21720 [*Penicillium rubens* Wisconsin 54–1255]
	*HypP*	−3.878218123	MFS transporter [*Purpureocillium lilacinum*]
	*MFSt*	−3.222620978	hypothetical protein NECHADRAFT [*Nectria haematococca*]

**Table 2 jof-07-01001-t002:** Thirty Candidate validation genes and reference gene Primer pair information.

Comparison Group	Primer Name	Primer Sequence 5′-3′
	GAPDH	ATGGAGAAGGCTGGGGCTCATTTGC; GATGACCCTTTTGGCTCCCCCCTGC
**CK-VS-T2**	Unigene0001578	ATGGAGAAGGCTGGGGCTCATTTGC; GATGACCCTTTTGGCTCCCCCCTGC
	Unigene0004220	ACGAACAGGATACAGGCGACA; CTGAGGTCCAAATAACGAGATGAA
	Unigene0004974	CAGGCGGTTTGGAACTCGT; TGCCATATCCATCCTGCGTC
	Unigene0005793	GCTACTGGCACGACATCGGAT; TCTAGTGGAGTGACAGCAAACGA
	Unigene0005693	ATGTCGGTGGCTGGGTTGA; CCCCTTTTGAGGCGCTGTAT
	Unigene0009373	TTGAGAATGTCTGGTTGCTGAATG; GCGCAGGTTGAGGAGTTGATGT
	Unigene0001633	CGGCAACTTCCCTGGTAGCT; GCCTGGCACGCTATACACCT
	Unigene0004295	TGCTGCTCTACCCTCGTTCTTG; CAATAGCCGTCTGGCAAAGGA
	Unigene0005088	GCTACAGATCGTCGCCTTGG; GCTAATACCGCATACGCCCTAC
	Unigene0008572	CCTACACTGGTCTCATCGGTTTC; GCTGGTAATCCTGCTCCATAAGA
**T1-VS-T2**	Unigene0001578	ATGGAGAAGGCTGGGGCTCATTTGC; GATGACCCTTTTGGCTCCCCCCTGC
	Unigene0004220	ACGAACAGGATACAGGCGACA; CTGAGGTCCAAATAACGAGATGAA
	Unigene0005793	GCTACTGGCACGACATCGGAT; TCTAGTGGAGTGACAGCAAACGA
	Unigene0001633	CGGCAACTTCCCTGGTAGCT; GCCTGGCACGCTATACACCT
	Unigene0005693	ATGTCGGTGGCTGGGTTGA; CCCCTTTTGAGGCGCTGTAT
	Unigene0009373	TTGAGAATGTCTGGTTGCTGAATG; GCGCAGGTTGAGGAGTTGATGT
	Unigene0001003	TCTAAACCCAACTCACGTACCACTT; AACAGGCTGATGACTCCCAAGA
	Unigene0005927	GGCTTTGCCATCCACCACA; GCAATGAAGAGGAGGTAGTCGC
	Unigene0007158	AGGATCGCAATGTGGAAGGC; GGTTGAGGGAAGAGTTTGAGGC
	Unigene0007851	GCAAGAGTCGCACGCAGAA; GCAGGTAAAGCCACAGTCCATAT
**CK-VS-T1**	Unigene0006182	TGCCATAACCCAATACCAAACG; TGTTGAGGTTGCGGTCTGTTAGT
	Unigene0000037	GAGTTCAGCCACGCCATCAC; CATTCCCGCTTCGACTTTGTT
	Unigene0000618	AGAGGAGAAGAACGCCGAGGT; CAATTTCGCCCACAAAGAGCT
	Unigene0000640	TCCAGGCAAGAACTACTACGGTC; CCTTCCAGAACCAGAAAGCAGT
	Unigene0000947	GCTTCTTCGCCTCCCACCTA; CGCTGTCTCGACAATGCACC
	Unigene0000948	ATGCTTTCAGCGTTTATCCCTT; CGTGAGACAGTTCGGTCCCTAT
	Unigene0005250	TTCCGCTTTAAGGTGTTCATGG; CAGGTCGCCGTAGTGGTTCA
	Unigene0006199	CTAGGCTCGTCACGTTTGCG; AACTGCTCCGGTTTGATGATATG
	Unigene0000783	TGGCTGATCAACTGACTGAGGAA; GCCCAGTTCCTTGGTGGTGAT
	Unigene0004166	CGAAGCGCCAAGATTAAGGA; CACTGCCCTCACCGTCAATG

**Table 3 jof-07-01001-t003:** *Ctf1β 1* and *Ctf1β 2* Gene knockout and complement experiment primer.

Primer Name	Primer Sequence 5′-3′
*Ctf1β1*-5-F	TAGGCCACCATGTTGGGCCCGCTGGCGTGTTGAGCATGA
*Ctf1β1-*5-R	AGTTCAGGCTTTTTCATATCTGGCAGGGCTGTTGGTAGA
*Ctf1β1-*3-F	CGAGGGCAAAGGAATAGAGTATGAAGCACGACCACCAGAT
*Ctf1β1-*3-R	GTGGACTCCTCTTAAAGCTTCGGCGAAAGTGAGTGGATT
*Ctf1β2-*5-F	TTCGGGTTACTTCCCTTCG
*Ctf1β2*-5-R	AGTTCAGGCTTTTTCATATCAGGACGCATTAGTCGAGGC
*Ctf1β2-*3-F	CGAGGGCAAAGGAATAGAGTTTAGGGCGTCATCTCGGTC
*Ctf1β2-*3-R	CCAGCATTCGTCAATATCAAGC
*Hph*-F	GATATGAAAAAGCCTGAACT
*Hph*-R	ACTCTATTCCTTTGCCCTCG
*Ctf1β1*-F	ATGGCCCCAAAAGACGGACTTTTGTCAG
*Ctf1β1*-R	GCGTACGAAGCTTCAGCTG TCAATGTGTGTGGCCAAGGCAAAGCT
*Ctf1β2*-F	ATGGCCGCAGACAACGAAGCTGT
*Ctf1β2*-R	GCGTACGAAGCTTCAGCTG TTACCCCTTTACCATACCCCCGTCC
KanMX-F	CAGCTGAAGCTTCGTACGC
KanMX-R	GCATAGGCCACTAGTGGATCTG
*Ctf1β1-*Up-F	GCTGGCGTGTTGAGCATGA
*Ctf1β1-Up*-R	CTGACAAAAGTCCGTCTTTTGGGGCCATTGGCAGGGCTGTTGGTAGA
*Ctf1β1-*Down-F	CAGATCCACTAGTGGCCTATGCATGAAGCACGACCACCAGAT
*Ctf1β1-*Down-R	CGGCGAAAGTGAGTGGATT
*Ctf1β2-*Up-F	TTCGGGTTACTTCCCTTCG
*Ctf1β2-*Up-R	ACAGCTTCGTTGTCTGCGGCCATAGGACGCATTAGTCGAGGC
*Ctf1β2-*Down-F	CAGATCCACTAGTGGCCTATGC TTAGGGCGTCATCTCGGTC
*Ctf1β2-*Down-R	CCAGCATTCGTCAATATCAAGC

Note: The sequence at the underline of the primer can be complementary paired with the sequences at both ends of the HPH fragment, and the red sequence can be complementary paired with the sequences at the vector enzyme digestion sites ApaI and Hind III.

**Table 4 jof-07-01001-t004:** Sequencing data statistics of the nine samples.

Sample	Clean Reads	GC Content	Reads Lenth	% ≥Q30
CK-1	52,922,016	56.50%	150	94.95%
CK-2	50,075,888	56.65%	150	94.92%
CK-3	48,964,936	56.68%	150	94.76%
T1-1	50,112,716	56.56%	150	94.87%
T1-2	58,737,286	56.47%	150	94.95%
T1-3	48,459,904	56.39%	150	94.92%
T2-1	50,093,080	56.47%	150	94.96%
T2-2	66,878,600	56.52%	150	94.76%
T2-3	59,806,800	56.46%	150	94.78%

Cultured in sterile deionized water medium for 7 days (CK); Cultured in sterile rice tissue liquid medium for 7 days (T1); Cultured in sterile *B. pervaiabilis × D. grandis* tissue fluid medium for 7 days (T2). Each treatment group had three duplications.

**Table 5 jof-07-01001-t005:** Statistical results of 9 samples compared with reference genes.

Sample	All Reads Number	Unique Mapped Reads	Multiple Mapped Reads	Mapping Ratio
CK-1	49,948,918	42,069,088	3,839,542	91.91%
CK-2	48,644,554	38,417,922	3,768,346	86.72%
CK-3	46,816,432	39,365,841	3,686,520	91.96%
T1-1	48,309,274	40,649,369	4,017,201	92.46%
T1-2	55,871,212	46,619,707	5,052,812	92.49%
T1-3	44,845,488	37,341,346	3,447,631	90.95%
T2-1	47,835,204	40,423,506	3,689,078	92.22%
T2-2	63,877,526	54,410,080	4,332,707	91.96%
T2-3	57,266,814	48,703,596	4,118,072	92.24%

## Data Availability

The raw sequence reads for this study were deposited into the NCBI Sequence Reads Archive (SRA) under the accession numbers SRR9278662, SRR9278661, SRR9278664, SRR9278663, SRR9278658, SRR9278657, SRR9278660, SRR9278659, SRR9278665. The assembled contigs has been published in the NCBI Transcriptome Shotgun Assembly (TSA) under the accession number GHWG00000000.

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
