# Peer review of "Comparative Transcriptomics and Gene Knockout Reveal Virulence Factors of Arthrinium phaeospermum in Bambusa pervariabilis × Dendrocalamopsis grandis"

_jof, 2021, doi:10.3390/jof7121001_

Round 1
Reviewer 1 Report
In the paper by Fang et al, the authors checked the differentially expressed genes in A. phaeospermum in different conditions include sterile deionized water (CK), rice tissue (T1) and hybrid bamboo (T2) fluid. The authors also checked the function of Ctf1β 1 and Ctf1β 2 genes by doing gene knockout study. This is really interesting piece of work. However, the authors should spend more time to revise the manuscript before resubmission. Please see my comments below:
- In section 2.1.1, the authors should bring some information on how they identified the isolated pathogen? Did they used sequencing for this purpose? If so, please bring accession number for that. Also you need to bring isolate culture collection number if you deposited the fungus in a culture collection.
- What was the purpose of using rice tissue extracts in your study as that part has nothing to do with the current research!
- In section 2.1.1, I wonder why the experiments happened in nature and not in the green house. Was it due to the fact that the bamboo plants are getting very tall? Please explain in more detail in the manuscript.
- I suggest to remove section 2.1.2 and bring the information wherever needed through the manuscript.
- In section 2.2.3, which kit did you use? You don’t need to bring the names of all reagents!
- In Line 174, please bring the reference for BLAST2Go software.
- Line 183, this is duplicate information that you already had in line 174!
- Line 190, the reference for RSEM software is missing.
- In section 2.2.6, how did you choose 30 candidate genes? Please bring some more information.
- In your experiments, did you have three biological replicates as well? If so, please add the information to the manuscript.
- Did you use any control for RT-PCR analysis? If so, please bring the information.
- In your RT-PCR analysis, how many housekeeping genes did you use? Was it only one? I suggest to have at least two housekeeping genes to have more accurate results.
- How did you design primers used in this research?
- In section 2.2.8, you need to bring the vector map and illustrate how you did the knock out construct. Illustration would add more value to this section and will make it more understandable for the readers.
- Please also explain in more detail, how you added homologous regions of hygromycin to the primers in section 2.2.8. Did you add them to the 5’ or 3’ end of the primers? Also explain if you check for promotor/terminator to amplify hygromycin? Which promotor did you use for that?
- In section 2.2.9, did you use freshly made protoplasts or did you save the protoplasts in -80 for future usage?
- In line 258, how much of protoplasts did you use? Also what was the final concentration of the protoplast in your experiemnts? Was it 107/ml in all cases?
- Line 258, what was the time intervals between each time you added PEG?
- In line 259, please bring the components of TB3 medium.
- In line 263, which lysing buffer did you use? Please bring the components on how you made it.
- Did you do any stability test for the mutants?
- Did you use any reference for disease index calculation? If so, please add the reference to the appropriate section.
- In section 2.1.12, please add more information in detail on how did you obtained the recombinant fragments?
- In line 294, did you use 30ul each for KanMx-Ctf1β 1-55/KanMx+Ctf1β 1-3 and KanMx-Ctf1β 2-55/KanMx-Ctf1β 2-3? Or 15ul each?
- In line 296, again, how much of protoplasts did you use?
- Again you need an illustration on how you did the complementation experiment?
- For the section 3.1.1, please bring the information on how did you do the statistical analysis? Also for figure 1. Maybe use different patterns so that the differences could be seen in easier way.
- In line 327, which reference sequence did you use? Please bring more information here.
- For figure 7, please add the statistical analysis on top of each bars.
- In lines 451-455, did you draw any kill curve for your statement? If not, I suggest to add your data as a kill curve for all used antibiotics in the supplementary files.
- In line 459, what is 15000 markers?
- In figure 12, please bring more information in the legend as to how to read letters? In another word, in each case, what was the comparisons?
- In figure 14, again, please use different patters as the current format is very unclear.
- Did you do any southern blot analysis to check the integration of the resistance marker to the genome? It is recommended to do so.
- I might missed it but did you do complemented studies for both knocked out genes?
- Scientific names of fungi/plants should be in italic all the time.
- The paper contains a lot of grammatical and language errors in the current format. Please ask a native speaker to read the paper before you send it back for review.
Reviewer 2 Report
The manuscript “Comparative transcriptomics and gene knockout reveal virulence factors of Arthrinium phaeospermum in Bambusa pervariabilis × Dendrocalamopsis grandis”, is an interesting paper that give some clues about Arthrinium phaeospermum mechanisms of pathogenicity. However, the authors design an experimental assay that did not compare the mycelial growth with in planta growth, which, in my point of view would have been much more interesting than the comparison between mycelial growth in PDA or PDA supplemented with Bambusa pervariabilis × Dendrocalamopsis grandis leaves or rice leaves.
The RNAseq analyses done between the 3 conditions is a loss of money and work once it only allows a partial view of the pathogenicity mechanisms. Another important issue is the molecular control along the infection process, that was not taken into account.
On the other side, the know-out experiments followed by the know-out complementation reveals the importance of cutinase transcription factor in this patho-system, open routes for novel and interesting studies.
Finally, for publication the manuscript needs an accurate revision to clean-up incautious errors.
Round 2
Reviewer 1 Report
- In section 2.1.8, the authors \should still bring some more information and answer the followings: How transformation to coli is done here? Which method was used for THIS PURPOSE in detail? Also, why the authors put the final fusion fragment in to P-Cambia vector? Were the authors put restriction enzymes to end of primers for PCR amplifications before putting the final fragments to the vector? If so, please bring related information in this section. Did the authors used same protocol as was recommended in the cloneexpress kit or they modified the protocol? Where is the reference for this protocol? Also please explain why you chosed Apaâ… 1μL, Hind â…¢ restriction enzymes specifically?
- Again, for making your complementation construct, did you use the same method as it shown for making your knock out construct? I think adding illustration on how you made complementation construct would add more value to this paper.
Reviewer 2 Report
After authors revisions lots of raised questions still without answer!
Such as:
Reference genes in qPCR assays;
De novo assembly versus mapping of reads in the available genome, at least a comparison with the genome;
Figure captions are still not self-explanatory;
Round 3
Reviewer 2 Report
Dear authors
My main concerns were appropriatly answered!
I still only without understand, despite the results did not change, why the RT-qPCR data were not normalised with 2 reference genes, once they performed the analysis, and this is a quality requirement for qPCR.

Author Response
Please see the attachment.

This manuscript is a resubmission of an earlier submission. The following is a list of the peer review reports and author responses from that submission.